# Study of reference intervals for free triiodothyronine, free thyroxine, and thyroid-stimulating hormone in an elderly Chinese Han population

**Jingting Xiong** [1☯], **Shiguo Liu** [2☯], **Kai Hu** [3☯], **Yinxiang Xiong** [1], **Pengyun Wang** [1]*, **Liang Xiong** [1]*

1 Department of Laboratory Medicine, Liyuan Hospital, Tongji Medical College, Huazhong University of Science and Technology, Hubei, PR China, 2 Department of Laboratory Medicine, Hubei No. 3 People's Hospital of Jianghan University, Hubei, PR China, 3 Department of Dermatological, Puai Hospital, Hubei, PR China

☯ These authors contributed equally to this work.
* xionglianghust@163.com (LX); wpy0110@126.com (PW)

**Data Availability Statement:** All relevant data are within the paper.

**Funding:** This work was supported by grants from the Training Plan of Young and Middle-Aged

## Abstract

The clinical manifestations of thyroid diseases in elderly patients are often atypical. This study aimed to establish reference intervals for thyroid function in the elderly in order to help diagnose thyroid diseases in this population. A total of 5345 healthy individuals were examined and divided into three groups according to their age: 4297 individuals aged < 65 years (19–64), 719 individuals aged between 65 and 79 years, and 329 individuals aged between 80 and 100 years. Levels of free triiodothyronine (FT3), free thyroxine (FT4), thyroid-stimulating hormone (TSH), thyroid peroxidase antibody, and thyroglobulin antibody were measured in these subjects by using a fully automated analyzer. The following free triiodothyronine, free thyroxine, and thyroid-stimulating hormone reference intervals were obtained from each age group: For individuals aged < 65 years (19–64 years), FT3, FT4, and TSH were 3.40–6.44, 10.26–19.25 pmol/L and 0.50–4.81 μIU/mL, respectively. For individuals aged between 65 and 79 years, FT3, FT4 and TSH ranged between 3.01–5.91, 10.04–19.76 pmol/L, and 0.54–5.51 μIU/mL, respectively. For individuals aged between 80 and 100 years, FT3, FT4, and TSH varied between 2.82–5.57, 9.79–21.22 pmol/L, 0.31–6.28 μIU/mL respectively. FT3 concentration was lower and the concentrations of FT4 and TSH were higher in individuals aged $\geq$ 65 years than in those aged <65 years (P<0.0001; P = 0.0039; P<0.0001, respectively). In conclusion, establishment of a reference interval would allow clinicians to diagnose diseases more accurately and easily.

## Introduction

Thyroid diseases are common endocrine diseases. Thyroid function testing is critical for the diagnosis of thyroid diseases [1]. Currently, the three most common tests for thyroid function

Backbone Talents of Medicine of Health and Family Planning Commission of Wuhan, Hubei Province, China (2017).

**Competing interests:** All authors declare no conflict of interest for this manuscript.

are the evaluation of thyroid-stimulating hormone (TSH), free thyroxine (FT4), and free triio-dothyronine (FT3) concentrations in the serum of venous blood. Studies revealed that different concentrations of iodine intake in different regions affect thyroid function [2–4]; thyroid function may also decrease with aging[5]. An inhibitory effect on the hypothalamic-pituitary axis, destructive thyroiditis, or immune-reactivating mechanisms can affect thyroid function [6].

As we grow older, the secretion, metabolism, and effects of thyroid hormones change. The hypothalamic-pituitary-thyroid axis maintains the thyroid function through complex regulatory mechanisms [7]. Researchers have recognized that the prevalence of thyroid disease increases with age [8]. The thyroid gland plays an important role in the aging process and in the endocrine system [9].

The prevalence of thyroid disease in the elderly population remains controversial. Herrmann reported that thyroid hormone secretion is reduced in healthy elderly people, decreased serum T3 concentration is not a result of low T3 syndrome, but of old age [10]; the serum T4 concentration is essentially unchanged [11]. There may be an increase in serum TSH concentrations [12]. A study has also shown the minimal impact of serum TSH concentrations on the diagnosis of thyroid dysfunction [13]. It is necessary to update the thyrotropin reference range for older patients [14]. The results of the various epidemiological studies conducted in different regions are closely related to the experimental methods used, such as previously used radioimmunoassay methods, the diagnostic criteria, and the different individuals diagnosed [1, 15, 16]. Therefore, there is a need to establish specific reference intervals for measuring thyroid function in the elderly patients in specific regions.

## Methods and materials

### Study participants and data collection

We conducted a retrospective study focusing on the elderly thyroid function and collected the thyroid function results of 2018 from the Laboratory Information System in 2019. Between Jan. 1, 2018 and Dec. 31, 2018, thyroid function test results of 12,670 individuals were collected from the Liyuan Hospital affiliated to Tongji Medical College of Huazhong University of Science and Technology, Zhongnan Hospital affiliated to Wuhan University Medical College, Hubei Third People's Hospital, and Wuhan Chinese Medicine Hospital. According to the criteria established by the American Association of Clinical Chemistry (AACC) Academy [17], the total number of individuals without thyroid dysfunction, visible or palpable goiter, treatment with drugs which can alter thyroid function test results such as glucocorticoids, adrenal insufficiency, renal insufficiency (failure) or other serious systemic diseases, or pregnancy was 10,446. Five tests for thyroid function (FT3, FT4,TSH, TPOAb, and TGAb) were conducted in venous blood samples from individuals in a fasting state (collected between 8:00-10:00 am), revealing that 5345 individuals, aged between 19 and 100 years, met the criteria of being negative for TPOAb and TGAb antibodies

The study was conducted in accordance with the Declaration of Helsinki (2008) and Liyuan Hospital, Tongji Medical College, Huazhong University of Science and Technology Institutional Review Board Approval [2018] IEC (A001). Informed consent was not required owing to the retrospective nature of the study.

### Laboratory measurements

The serum concentrations of FT3, FT4, TSH, TPOAb, and TGAb were detected using the LIAISON XL fully automated chemiluminescence immunoassay analyzer (LIAISON XL, DiaSorin, Italy) and its supporting reagents and calibrators. Each specimen was tested once within

the normal quality control range. The reference intervals for TPOAb and TGAb were in accordance with the LIAISON reagent instructions for TPOAb (1–16 IU/mL) and TGAb (0–100 IU/mL).

### Statistical analysis

Statistical analyses were conducted using SPSS statistical software, version 16.0 (Chicago, IL, USA) and GraphPad Prism, version 6.0 (GraphPad Prism Software, San Diego, CA, USA). Kolmogorov-Smirnov tests were used to evaluate the distribution of data; normally distributed data are presented as mean ± standard deviation (SD), whereas non-normally distributed data were calculated by logarithmic transformation. The reference range was defined as the 2.5th and 97.5th percentile (P2.5–P97.5), using SPSS. A P-value < 0.05 was considered statistically significant.

## Results

### Normality analysis of individuals in the three age groups

Three groups were analyzed, in which individuals were aged < 65 years (19–64 group), between 65 and 79 years (65–79 group), and between 80 and 100 years (80–100 group). In the < 65 group, FT3 followed an approximately normal distribution (Fig 1A), whereas FT4 and TSH were abnormally distributed (they were then logarithmically transformed, as shown in Fig 1B and 1C). In the 65–79 group, FT3 followed an approximately normal distribution (Fig 1D), whereas FT4 and TSH were abnormally distributed (they were then logarithmically transformed, as shown in Fig 1E and 1F). In the 80–100 group, FT3, FT4, and TSH were all abnormally distributed (they were then logarithmically transformed, as shown in Fig 1G, 1H and 1I). For the processing of some abnormally distributed data, we employed the logarithmic transformation $x^* = \log(x + 1)$, where $x^*$ was approximated in accordance with the normal distribution; the resulting frequency histograms shown in Fig 1 were then analyzed.

### Major thyroid function analysis

The reference intervals for each age group, after the correction for normality, are shown in Table 1.

### Analysis of the differences in thyroid functions

Two groups (individuals aged < 65 years [19–64 group], and ≥ 65 years [the 65+ group]) were analyzed using an independent *t*-test. The results of the analysis are presented in Fig 2.

The concentration of FT3 in the 65+ group was lower than that of the < 65 group (P < 0.0001). The concentration of FT4 in the 65+ group (P = 0.0039), and the TSH concentration in the 65+ group (P < 0.0001) were both higher than that of the < 65 group.

Two groups (individuals aged 65–79 years and 80–100 years) were analyzed using an independent *t*-test. The results of the analysis are presented in Fig 3.

The concentration of FT3 was lower (P < 0.0001), and the concentration of FT4 was higher (P = 0.0012) in individuals aged 80–100 years than those of the individuals aged 65–79 years. The TSH concentration was not significantly different between the two groups (P = 0.4439).

## Discussion

As elderly individuals do not present obvious signs and symptoms to enable easy diagnosis of thyroid-related diseases, it is difficult to diagnose diseases such as hyperthyroidism. This may lead to delayed treatment or even more serious consequences for the elderly [18]. Given the age of the population, the significance of thyroid dysfunction in elderly individuals remains

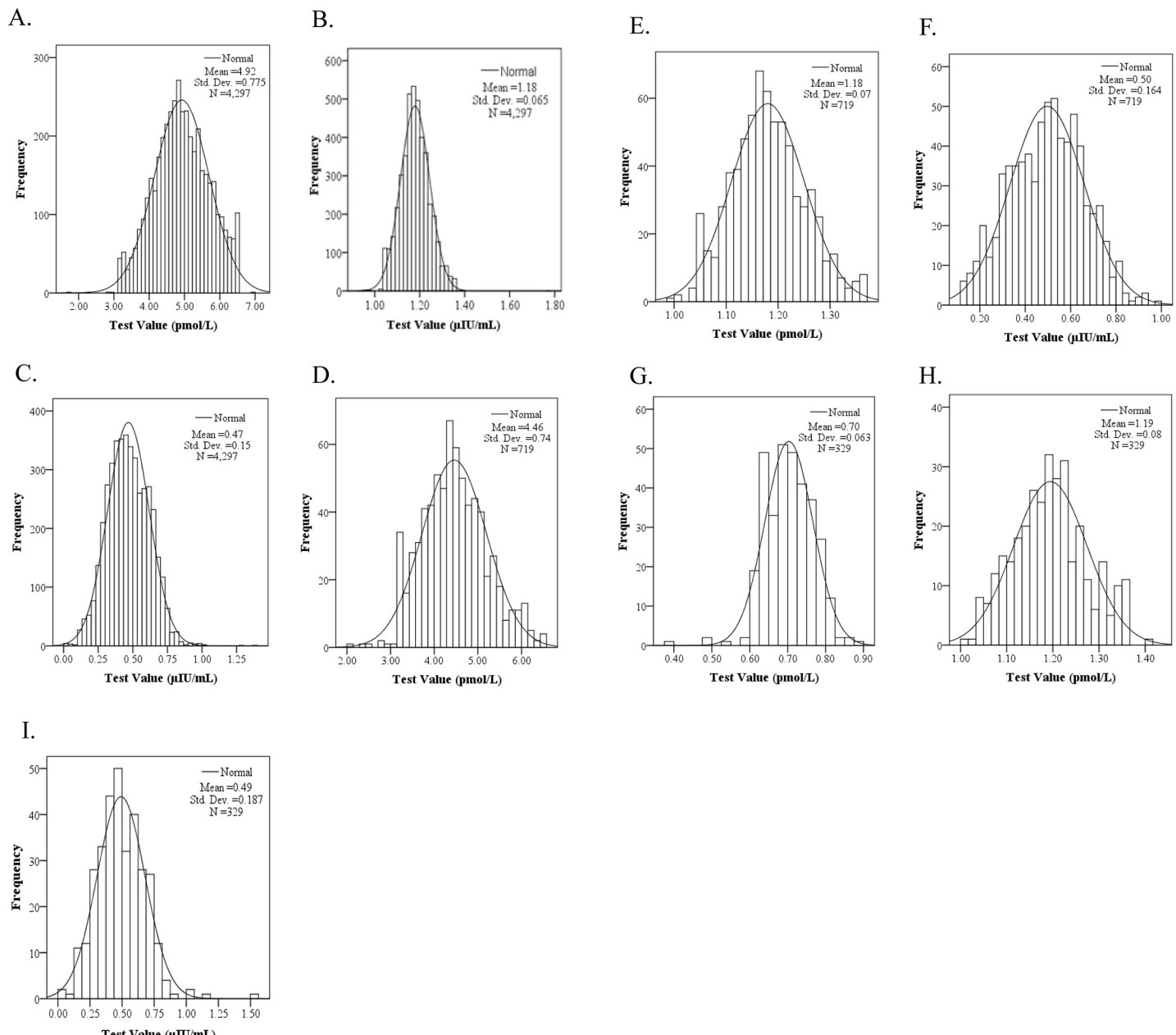

**Fig 1. Normal distribution of FT3, FT4, and TSH in different age groups.**

**Table 1. Reference intervals for the major thyroid functions in the elderly and non-elderly.**

| | FT3 (pmol/L) | | | FT4 (pmol/L) | | | TSH (µIU/mL) | | |
|---|---|---|---|---|---|---|---|---|---|
| | x/x* | SD/SD* | Reference Interval | x/x* | SD/SD* | Reference Interval | x* | SD* | Reference Interval |
| < 65 | 4.92 | 0.78 | 3.40–6.44 | 1.18 | 0.065 | 10.26–19.25 | 0.47 | 0.15 | 0.50–4.81 |
| 65–79 | 4.46 | 0.74 | 3.01–5.91 | 1.18 | 0.07 | 10.04–19.76 | 0.5 | 0.16 | 0.54–5.51 |
| 80–100 | 0.70 | 0.06 | 2.82–5.57 | 1.19 | 0.08 | 9.79–21.22 | 0.49 | 0.19 | 0.31–6.28 |

x* was obtained from the logarithmic conversion of x, x* = log(x + 1). SD, standard deviation. SD* was measured from x*.

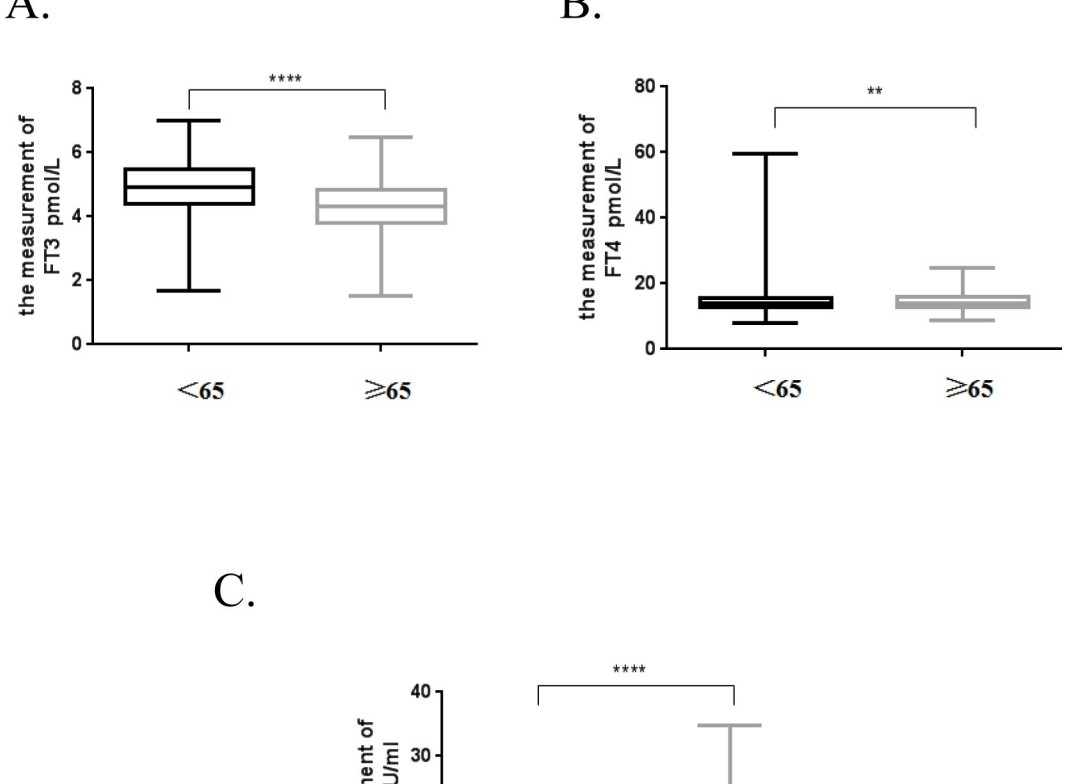

**Fig 2. Analysis of the differences in major thyroid functions between the two groups.**<

insufficiently understood. Mariotti et al. found that serum TSH, FT3 and rT3 concentrations are lower, higher and higher, respectively in centenarians than in younger individuals, whereas the FT4 concentration remained in the normal range [19]. With aging, the secretion of many hormones (such as estrogen, testosterone, and growth hormones) and hormone-sensitivity of tissues are reduced [20]. Therefore, common reference intervals for adult and elderly populations can lead to a misdiagnosis of thyroid diseases, and it becomes necessary to use an age-specific reference interval for TSH in the elderly population. A precise reference interval can avoid the diagnosis of thyroid dysfunction caused by the misclassification of "subclinical" thyroid disease [21,22]. In elderly individuals, mild disorders of thyroid function are currently less well recognized. Studies have shown that in regions with adequate iodine intake, serum TSH concentrations increase with age. Cross-sectional studies have shown that the concentration of FT4 is normal or slightly reduced in elderly individuals; other studies suggest that FT4 is slightly or not obviously increased in elderly individuals. Most studies suggest that serum FT3 concentrations decrease with age [23].

When measured by an electrochemiluminescence immunoassay method using a Cobas 601 analyzer (Roche Diagnostics, Switzerland) in regions of China in 2017, and when the reference range for the general population was used, it was found that the prevalence of hypothyroidism

A.

B.

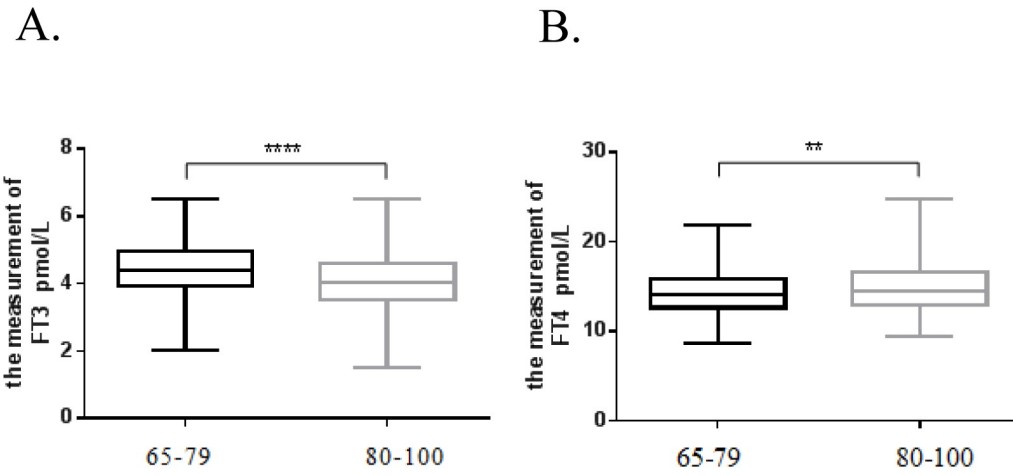

C.

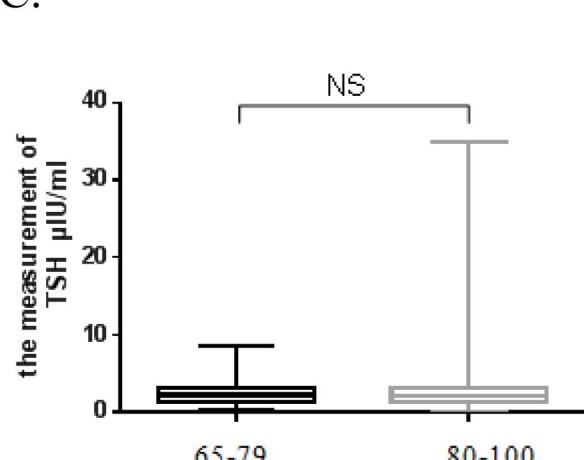

**Fig 3. Analysis of the differences in major thyroid functions in elderly individuals in two age groups (65–79 years and 80–100 years).**

and subclinical hypothyroidism in individuals aged $\geq$ 65 years was significantly higher than that of individuals aged < 65 years. When using age-specific reference ranges, the prevalence of hypothyroidism in individuals aged $\geq$ 65 years was lower than that of the general population [24]. There are certain difficulties in standardization and consistency owing to the different detection methods (the use of different antibodies); the differences between the detection results are large, and the results are difficult to compare. Reference intervals for different immune detection platforms need to be created. For detection, we used the DiaSorin LIAISON XL analyzer, which uses an isoluminol derivative as the chemiluminescent substrate. Therefore, the same reference range cannot be used on the Cobas 601 analyzer.

For this study, we selected the test population according to the standards of the AACC Academy to detect the main thyroid function in the Chinese Han population. The results of

this analysis grouped selected healthy people by age, into individuals aged < 65 years, 65–79 years, and 80–100 years. We found that the FT3 concentration in the Chinese Han population was lower in individuals aged < 65 years than in those aged ≥ 65 years. Moreover, the same tendency was found for FT3 and FT4 concentrations in the groups of individuals aged 65–79 years and 80–100 years. This phenomenon was consistent with some previous studies; therefore, establishing reference intervals for the main thyroid function in elderly individuals from this area satisfies a clinical need. Essentially, the concentration of FT3 decreased, and the concentration of FT4 increased with age.

Overall, this study focused on the retrospective analysis of thyroid function in an elderly Chinese Han population. To achieve better monitoring of the thyroid hormones concentrations in elderly Chinese individuals, in accordance with the requirements of the CLSI C28-A3 document, statistical analysis of data from more regions and more centers is needed.

## Author Contributions

**Conceptualization:** Jingting Xiong, Liang Xiong.

**Data curation:** Jingting Xiong, Shiguo Liu, Yinxiang Xiong, Liang Xiong.

**Formal analysis:** Jingting Xiong, Kai Hu.

**Funding acquisition:** Pengyun Wang.

**Investigation:** Yinxiang Xiong, Pengyun Wang.

**Methodology:** Liang Xiong.

**Project administration:** Liang Xiong.

**Resources:** Shiguo Liu.

**Software:** Jingting Xiong, Kai Hu.

**Supervision:** Kai Hu, Pengyun Wang, Liang Xiong.

**Validation:** Jingting Xiong, Pengyun Wang, Liang Xiong.

**Visualization:** Pengyun Wang, Liang Xiong.

**Writing – original draft:** Jingting Xiong.

**Writing – review & editing:** Jingting Xiong, Shiguo Liu, Kai Hu, Yinxiang Xiong, Pengyun Wang, Liang Xiong.

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
