## [Decision Letter · Decision Letter 0]

11 Nov 2019

PONE-D-19-14834

Study of reference intervals for main thyroid functions in an elderly Chinese Han population using fully automated chemiluminescence immunoassays

PLOS ONE

Dear Mrs xiong,

Thank you for submitting your manuscript to PLOS ONE. After careful consideration, we feel that it has merit but does not fully meet PLOS ONE’s publication criteria as it currently stands. Therefore, we invite you to submit a revised version of the manuscript that addresses the points raised during the review process.

We would appreciate receiving your revised manuscript by Dec 26 2019 11:59PM. To enhance the reproducibility of your results, we recommend that if applicable you deposit your laboratory protocols in protocols.io, where a protocol can be assigned its own identifier (DOI) such that it can be cited independently in the future. For instructions see: http://journals.plos.org/plosone/s/submission-guidelines#loc-laboratory-protocols

We look forward to receiving your revised manuscript.

Kind regards,

Marcello Ciaccio, M.D., Ph.D

Academic Editor

PLOS ONE

Journal Requirements:

1. Thank you for including your ethics statement:   "The study was conducted in accordance with the Declaration of Helsinki (2008) and Liyuan Hospital, TongjiMedical College，Huazhong University of Science and Technology Institutional Review Board Approval.[2018] IEC (A001)."

Please amend your current ethics statement to confirm that your named institutional review board or ethics committee specifically approved this study.

2. Please amend the manuscript submission data (via Edit Submission) to include authors Shiguo Liu, Kai Hu, Yinxiang Xionga, Pengyun Wanga, Liang Xionga

Reviewers' comments:

Reviewer's Responses to Questions

**Comments to the Author**

1. Is the manuscript technically sound, and do the data support the conclusions?

Reviewer #1: Yes

2. Has the statistical analysis been performed appropriately and rigorously? 

Reviewer #1: Yes

3. Have the authors made all data underlying the findings in their manuscript fully available?

Reviewer #1: Yes

4. Is the manuscript presented in an intelligible fashion and written in standard English?

Reviewer #1: Yes

5. Review Comments to the Author

Reviewer #1: There are two major concerns in the design of the study.

Please provide the ultrasound of the recruited cases, as well as TRAb results. These two parameters are also important to determine whether the included cases are with normal thyroid or not, especially thyroid ultrasound. Visible and palpable goiter is not enough.

Please describe why chose LIAISON to test thyroid function. It is better to use more than one method (at least two) to measure thyroid function.

6. PLOS authors have the option to publish the peer review history of their article (what does this mean?). If published, this will include your full peer review and any attached files.

Reviewer #1: Yes: Zhaowei Meng

---

## [Author Response · Author response to Decision Letter 0]

1 Dec 2019

Dear Editors and Reviewers:

Thank you for your letter and for the reviewer’s comments concerning our manuscript entitled “Study of reference intervals for main thyroid functions in an elderly Chinese Han population using fully automated chemiluminescence immunoassays” [manuscript number: PONE-D-19-14834]. The comments were all very helpful for revising and improving our manuscript, as well as for providing significant guidance to the researches of our laboratory. We have amended ethics statement statements in the Methods section of the manuscript. We have studied all comments carefully and have made corrections in the manuscript accordingly. We hope that the revised manuscript meets your expectations. The revised portions are marked in red in the manuscript. The main corrections in the paper and the responses to the reviewer’s comments are given below.

Response to the reviewer’s comments:

1.Comment: Please provide the ultrasound of the recruited cases, as well as TRAb results. These two parameters are also important to determine whether the included cases are with normal thyroid or not, especially thyroid ultrasound. Visible and palpable goiter is not enough.

Response: Thank you for your constructive advice. We agree with you that the above-mentioned two parameters are important to determine whether the included cases had a normal thyroid or not. However, an ultrasound test is not routinely performed for every hospitalized patient and during each health checkup because of financial reasons, among others. Ours was a retrospective study. We excluded cases that showed obvious goiter upon visual inspection and palpation by a doctor; an ultrasound test was carried out for cases where goiter could not be excluded by palpation. A similar procedure was followed in a previous research[1]. 

 TRAb tests for antibodies, such as TPOAb, TGAb and TRAb, are used for identifying autoimmune thyroid conditions widely. However, TRAb tests are not widely used for every hospitalized patient and during each health checkup because of its controversial clinical applicability[2]. According to the criteria established by the American Association of Clinical Chemistry (AACC) Academy[2], TSH reference intervals should be established from persons who are not with detectable thyroid autoantibodies, TPOAb or TgAb. Some other studies that have established RIs for thyroid hormones also used TPOAb and TGAb[1, 3]. 

2.Comment: Please describe why chose LIAISON to test thyroid function. It is better to use more than one method (at least two) to measure thyroid function.

 Response: Thank you for your kind advice. As is shown in the study of reference intervals for thyroid function, Roche, Abbott, Siemens, and Beckman analyzers have been used for developing reference intervals for thyroid function[1, 4-6]; the LIAISON analyzer also employs fully automated chemiluminescence and performs stably in daily use[7]. Reference intervals need to be set separately for different immune detection platforms used in special projects. There are still many hospitals in China that use the LIAISON analyzer to perform thyroid tests for samples daily. It is essential for clinical laboratories to establish reference intervals for the LIAISON analyzer for regular health checkups or evaluation of patients who visit hospitals. 

 There are three methodologies (RIA, IMA, and LC-MS/MS) currently being used for the thyroid function test. The use of radioimmunoassay (RIA) is limited due to the resulting radioactive contamination. Liquid chromatography-tandem mass spectrometry (LC-MS/MS) has progressively improved the specificity of the thyroid function test, but its use is also limited because of the requirement of expensive manually operated machinery. Currently, thyroid testing is performed on serum specimens using fully automated chemiluminescence immunoassays in most clinical laboratories. Until between-method biases are eliminated, it is not feasible to propose universal reference ranges that would apply across methods. 

Thank you very much for considering this work. We look forward to hearing from you soon.

If you have any questions, please feel free to contact us.

Best regards

Sincerely,

Liang Xiong

Department of Laboratory Medicine, Liyuan Hospital, Tongji Medical College, Huazhong University of Science and Technology, Yanhu Road, 39 Wuhan, Hubei, PR China

Phone number: 86-027-86785006

Fax number: 86-027-86793043

xionglianghust@163.com

[1] Clerico A, Trenti T, Aloe R, Dittadi R, Rizzardi S, Migliardi M et al. A multicenter study for the evaluation of the reference interval for TSH in Italy (ELAS TSH Italian Study)[J]. Clin Chem Lab Med. 2018, 57: 259-267.

[2] Demers LM, Spencer CA. Laboratory medicine practice guidelines: laboratory support for the diagnosis and monitoring of thyroid disease[J]. Clin Endocrinol (Oxf). 2003, 58: 138-140.

[3] Zhai X, Zhang L, Chen L, Lian X, Liu C, Shi B et al. An Age-Specific Serum Thyrotropin Reference Range for the Diagnosis of Thyroid Diseases in Older Adults: A Cross-Sectional Survey in China[J]. Thyroid. 2018, 28: 1571-1579

[4] Ehrenkranz J, Bach PR, Snow G L, Schneider A, Lee JL, Ilstrup S et al. Circadian and Circannual Rhythms in Thyroid Hormones: Determining the TSH and Free T4 Reference Intervals Based Upon Time of Day, Age, and Sex[J]. Thyroid. 2015, 25: 954-961.

[5] Sriphrapradang C, Pavarangkoon S, Jongjaroenprasert W, Chailurkit L-o, Ongphiphadhanakul B, Aekplakorn W. Reference ranges of serum TSH, FT4 and thyroid autoantibodies in the Thai population: the national health examination survey[J]. Clin Endocrinol (Oxf). 2014, 80: 751-756.

[6] Amouzegar A, Delshad H, Mehran L, Tohidi M, Khafaji F, Azizi F. Reference limit of thyrotropin (TSH) and free thyroxine (FT4) in thyroperoxidase positive and negative subjects: a population based study[J]. J Endocrinol Invest. 2013, 36: 950-954.

[7] Bulur O, Atak Z, Ertugrul D T, Beyan E, Gunakan E, Karakaya S et al. Trimester-specific reference intervals of thyroid function tests in Turkish pregnants[J]. Gynecol Endocrinol. 2019.

---

## [Decision Letter · Decision Letter 1]

28 Feb 2020

PONE-D-19-14834R1

Study of reference intervals for main thyroid functions in an elderly Chinese Han population using fully automated chemiluminescence immunoassays

PLOS ONE

Dear Dr. Xiong,

Thank you for submitting your manuscript to PLOS ONE. After careful consideration, we feel that it has merit but does not fully meet PLOS ONE’s publication criteria as it currently stands. Therefore, we invite you to submit a revised version of the manuscript that addresses the points raised during the review process.

We would appreciate receiving your revised manuscript by Apr 13 2020 11:59PM. To enhance the reproducibility of your results, we recommend that if applicable you deposit your laboratory protocols in protocols.io, where a protocol can be assigned its own identifier (DOI) such that it can be cited independently in the future. For instructions see: http://journals.plos.org/plosone/s/submission-guidelines#loc-laboratory-protocols

We look forward to receiving your revised manuscript.

Kind regards,

Silvia Naitza

Academic Editor

PLOS ONE

Additional Editor Comments (if provided):

Dear Dr. Xiong,

sorry for the delay in the review process of your manuscript, due mainly to the difficulties in finding a second Reviewer for your work. We have now received two independent reviews and as you will see from the attached comments, we cannot accept your manuscript as it stands now. Please, address all the points raised by Reviewer 2 and submitt a revised version of your manuscript including a point-by-point response.

Best regards,

Silvia Naitza

Reviewers' comments:

Reviewer's Responses to Questions

**Comments to the Author**

1. If the authors have adequately addressed your comments raised in a previous round of review and you feel that this manuscript is now acceptable for publication, you may indicate that here to bypass the “Comments to the Author” section, enter your conflict of interest statement in the “Confidential to Editor” section, and submit your "Accept" recommendation.

Reviewer #1: All comments have been addressed

Reviewer #2: (No Response)

2. Is the manuscript technically sound, and do the data support the conclusions?

Reviewer #1: Yes

Reviewer #2: No

3. Has the statistical analysis been performed appropriately and rigorously? 

Reviewer #1: Yes

Reviewer #2: No

4. Have the authors made all data underlying the findings in their manuscript fully available?

Reviewer #1: Yes

Reviewer #2: (No Response)

5. Is the manuscript presented in an intelligible fashion and written in standard English?

Reviewer #1: Yes

Reviewer #2: Yes

6. Review Comments to the Author

Reviewer #1: This paper intends to sstablish a reference interval which allows clinicians to diagnose diseases more accurately and easily. All questions are answered. This paper can be accepted.

Reviewer #2: This paper describes thyroid function test results obtained from a single centre (hospital) in China. The authors have arranged TFT results by age groups and compiled reference ranges.

This reviewer agrees that thyroid function does change with age and having age-specific reference ranges is important in ensuring that incorrect diagnosis is not made and unnecessary treatment not initiated.

This paper needs to address several issues.

1. Thyroid function - especially TSH and FT3 levels - are affected by a number of factors including drugs and medical illnesses. It isn't clear if the patients that were included were in-patients or out-patients. Older people and those that already have underlying medical conditions are more likely to be admitted to hospital. Thus, changes in thyroid function may reflect their health rather than be due to their age per se. Similarly, a number of drugs such as amiodarone, lithium, metformin and anti-convulsants among others can affect thyroid function. This needs to be clarified.

2. Thyroid function particularly TSH and FT3 levels have a circadian rhythm and can be affected by food. Were the samples obtained at different times of the day and night and in fasting and non-fasting states?

3. Title and abstract: not sure what "main" thyroid functions are?

4. Abstract: please explain how the reference intervals were calculated. This is important as this is the main focus of the analysis.

5. Introduction (end of first para): several factors affect thyroid function and not just iodine intake and ageing. Please see Jonklaas and Razvi. Lancet Diab Endocrinology 2019 PMID: 30797750

6. Introduction (second para): "Researchers have connected aging with hypothyroidism

and believe that hypothyroidism is a manifestation of aging". Please revise this sentence or provide a reference.

7. Informed consent was not required as this was a retrospective study. Consent has to do with using identifiable information being used with the patient's knowledge and not with the design of the study. Maybe there was no identifiable information available to reserchers and that is why no consent was required?

8. Statistical analysis: this is one of the shortest statistical analysis section I have ever seen. Please provide details of how reference ranges were calculated. And how data are presented.

9. Typo on page 10: do you mean rT3?

10. Please include a reference to the recent editorial by Cappola A in JAMA in 2019 The Thyrotropin reference range should be changed in older people. PMID: 31664455

I hope you find these comments constructive.

7. PLOS authors have the option to publish the peer review history of their article (what does this mean?). If published, this will include your full peer review and any attached files.

Reviewer #1: No

Reviewer #2: No

---

## [Author Response · Author response to Decision Letter 1]

19 Mar 2020

Dear Editor and Reviewers:

 Thank you for your letter and for the reviewers' comments concerning our manuscript entitled "Study of reference intervals for main thyroid functions in an elderly Chinese Han population using fully automated chemiluminescence immunoassays" [manuscript number: PONE-D-19-14834R1]. 

 The comments were extremely helpful for the revision and improvement of our manuscript, as well as for providing significant guidance to our research. We have studied all comments carefully and have made corrections in the manuscript accordingly. We hope that the revised manuscript meets your expectations. The revised sections are marked in red in the manuscript. 

 The major corrections in the paper and our responses to the reviewer's comments are given below.

Response to the reviewers' comments:

 1.Comment: Thyroid function - especially TSH and FT3 levels - are affected by a number of factors including drugs and medical illnesses. It isn't clear if the patients that were included were in-patients or out-patients. Older people and those that already have underlying medical conditions are more likely to be admitted to hospital. Thus, changes in thyroid function may reflect their health rather than be due to their age per se. Similarly, a number of drugs such as amiodarone, lithium, metformin and anti-convulsants among others can affect thyroid function. This needs to be clarified.

 Response: We agree that a number of factors, including drugs and medical illnesses, affect thyroid function. In this study, all the samples were obtained from in-patients, out-patients, and people who had a checkup done. The excluded ones had thyroid dysfunction, visible or palpable goiter, adrenal insufficiency, renal insufficiency (failure), or other serious systemic diseases; pregnant women or individuals taking drugs that can alter thyroid function test results such as glucocorticoids were also excluded. This has been revised in the Methods and materials section on page 4.

 2.Comment: Thyroid function particularly TSH and FT3 levels have a circadian rhythm and can be affected by food. Were the samples obtained at different times of the day and night and in fasting and non-fasting states?

 Response: Thank you for your kind remind. Fasting venous blood samples were collected between 8:00 and 10:00 am. We have added this in the section ‘Subjects and sample collection’ on page 5.

 3. Comment: Title and abstract: not sure what "main" thyroid functions are?

 Response: We have revised the mention of main thyroid functions for free triiodothyronine, free thyroxine and thyroid-stimulating hormone in the title and abstract. We changed the title to "Study of reference intervals for free triiodothyronine, free thyroxine, and thyroid-stimulating hormone in an elderly Chinese Han population".

 4. Comment: Abstract: please explain how the reference intervals were calculated. This is important as this is the main focus of the analysis.

 Response: The reference intervals were established using the CLSI C28-A3 document: Defining, Establishing, and Verifying Reference Intervals in the Clinical

Laboratory. According to this, the reference ranges of FT3/FT4/TSH were defined by the 2.5th and 97.5th percentile.

 5. Comment: Introduction (end of first para): several factors affect thyroid function and not just iodine intake and ageing. Please see Jonklaas and Razvi. Lancet Diab Endocrinology 2019 PMID: 30797750

 Response: It is true that not only iodine intake and ageing but also an inhibitory effect on the hypothalamic-pituitary axis, destructive thyroiditis, or immune-reactivating mechanisms, affect thyroid function. We have made the correction in the Introduction on page 3.

 6. Comment: Introduction (second para): "Researchers have connected aging with hypothyroidism and believe that hypothyroidism is a manifestation of aging". Please revise this sentence or provide a reference. 

 Response: We revised the sentence "Researchers have recognized that the prevalence of thyroid disease increases with age" and added the following reference: Arch Intern Med. 2000;160:526-534, PMID10695693.

 7. Comment: Informed consent was not required as this was a retrospective study. Consent has to do with using identifiable information being used with the patient's knowledge and not with the design of the study. Maybe there was no identifiable information available to reserchers and that is why no consent was required? 

 Response: Yes, informed consent was not required as this was a retrospective study. We obtained the necessary information from clinical and demographic information available on the Laboratory Information System (LIS). It can be found in Table 1: guidance on informed consent and ethical approval requirements based on the study type in the field of laboratory medicine. ( https://doi.org/10.11613/BM.2018.030201 PMID: 30429665 )

 8. Comment: Statistical analysis: this is one of the shortest statistical analysis section I have ever seen. Please provide details of how reference ranges were calculated. And how data are presented.

 Response: We have made changes as follows: Statistical analyses were conducted using SPSS statistical software, version 16.0 (Chicago, IL, USA) and GraphPad Prism, version 6.0 (GraphPad Prism Software, San Diego, CA, USA). Kolmogorov-Smirnov tests were used to evaluate the distribution of data; normally distributed data were presented as mean±SD, whereas non-normally distributed data were calculated by the logarithmic transformation. The reference range was defined by the 2.5th and 97.5th percentile (P2.5–P97.5), using SPSS. P-value < 0.05 was considered statistically significant.

 9. Comment: Typo on page 10: do you mean rT3?

 Response: We apologize for the error. We have changed γT3 to rT3.

 10. Comment: Please include a reference to the recent editorial by Cappola A in JAMA in 2019 The Thyrotropin reference range should be changed in older people. PMID: 31664455 

 Response: We added the sentence "It is necessary to update the thyrotropin reference range for older patients" with the reference on page 4 (Introduction, third paragraph).

 Thank you for your consideration. We look forward to hearing from you soon.

 If you have any questions, please feel free to contact us.

Sincerely,

Liang Xiong

Department of Laboratory Medicine, Liyuan Hospital, Tongji Medical College, Huazhong University of Science and Technology, Yanhu Road, 39 Wuhan, Hubei, PR China

Phone number: 86-027-86785006

Fax number: 86-027-86793043

Email: xionglianghust@163.com

---

## [Decision Letter · Decision Letter 2]

6 Aug 2020

PONE-D-19-14834R2

Study of reference intervals for free triiodothyronine, free thyroxine, and thyroid-stimulating hormone in an elderly Chinese Han population

PLOS ONE

Dear Dr. Jingting Xiong,

Thank you for submitting your manuscript to PLOS ONE. After careful consideration, we feel that it has merit but does not fully meet PLOS ONE’s publication criteria as it currently stands. Therefore, we invite you to submit a revised version of the manuscript that addresses the points raised during the review process.

In particular, we ask you to further improve the description of your study and describe more thoroughly the details of its retrospective nature, including when the retrospectives searches took place and the date ranges applied when you selected the eligible records. Furthermore, we kindly ask you to copyedit the manuscript in order to improve the English and gain in clarity. Please also note that we have included a copy of the two reviews on your revised version 2 of the manuscript, so please include the changes suggested by the Referee. 

We look forward to receiving your revised manuscript.

Kind regards,

Silvia Naitza

Academic Editor

PLOS ONE

Additional Editor Comments (if provided):

Dear Dr. Xiong,

first of all I'd like to apologize for the long time it took us to complete the review process of your manuscript PONE-D-19-14834R2, due to the need to clarify with the Journal Editorial Board the retrospective nature of your study and its approval by an Ethics committee, which was also a major concern of one of the Reviewers. Thank you for sending all the information and documentation requested. At this point we feel that this manuscript can proceed its review process. In particular, in order to make it suitable for publication, we ask you to further improve the description of your study and specifically describe the details of the retrospective nature more thoroughly, including when the retrospectives searches took place and the date ranges applied when you selected the eligible records. Furthermore, we feel that the manuscript should be copyedited to improve the English, in order to gain clarity and avoiding confusion in understanding. Please also note that we have included a copy of the reviews on your Revision 2 of the manuscript, so please include the changes suggested by the Referee.

Best regards,

Silvia Naitza

Reviewers' comments:

Reviewer's Responses to Questions

**Comments to the Author**

1. If the authors have adequately addressed your comments raised in a previous round of review and you feel that this manuscript is now acceptable for publication, you may indicate that here to bypass the “Comments to the Author” section, enter your conflict of interest statement in the “Confidential to Editor” section, and submit your "Accept" recommendation.

Reviewer #1: All comments have been addressed

Reviewer #2: All comments have been addressed

2. Is the manuscript technically sound, and do the data support the conclusions?

Reviewer #1: Yes

Reviewer #2: Partly

3. Has the statistical analysis been performed appropriately and rigorously? 

Reviewer #1: Yes

Reviewer #2: Yes

4. Have the authors made all data underlying the findings in their manuscript fully available?

Reviewer #1: Yes

Reviewer #2: No

5. Is the manuscript presented in an intelligible fashion and written in standard English?

Reviewer #1: Yes

Reviewer #2: Yes

6. Review Comments to the Author

Reviewer #1: This study aimed to establish reference intervals for thyroid function in the elderly. FT3 concentration was lower in individuals aged ≥ 65 years than in individuals aged < 65 years, whereas the concentrations of FT4 and TSH, were higher in individuals aged ≥ 65 years than in individuals aged < 65 years.

The revision is suitable for publication.

Reviewer #2: Thank you for the responses to my queries. The manuscript would benefit from clarity if more details of how patients were identified and their data was collected. Are all patients asked to attend in a fasting state and between 0800 and 1000 hours or were data collected only for participants who fitted this criteria?

7. PLOS authors have the option to publish the peer review history of their article (what does this mean?). If published, this will include your full peer review and any attached files.

Reviewer #1: **Yes: **Zhaowei Meng

Reviewer #2: No

---

## [Author Response · Author response to Decision Letter 2]

23 Aug 2020

August 23, 2020

Silvia Naitza

Academic Editor

PLOS ONE

Dear Editor and Reviewers:

Thank you for your letter and for the reviewers' comments concerning our manuscript entitled "Study of reference intervals for free triiodothyronine, free thyroxine, and thyroid-stimulating hormone in an elderly Chinese Han population" [manuscript number: PONE-D-19-14834R2]. 

The comments were extremely helpful for the revision and improvement of our manuscript. We have studied all comments carefully and have made corrections in the manuscript accordingly. We hope that the revised manuscript meets your expectations. The revised sections are marked in red in the manuscript. 

1.Comment: we ask you to further improve the description of your study and describe more thoroughly the details of its retrospective nature, including when the retrospectives searches took place and the date ranges applied when you selected the eligible records. 

Response: We have improve the description in the section ‘Study participants and data collection’on page 4-5 as follows: We conducted a retrospective study focusing on the elderly thyroid function and collected the thyroid function results of 2018 from the Laboratory Information System in 2019. Between Jan. 1, 2018 and Dec. 31, 2018, thyroid function test results of 12,670 individuals were collected from the Liyuan Hospital affiliated to Tongji Medical College of Huazhong University of Science and Technology, Zhongnan Hospital affiliated to Wuhan University Medical College, Hubei Third People’s Hospital, and Wuhan Chinese Medicine Hospital. According to the criteria established by the American Association of Clinical Chemistry (AACC) Academy [17], the total number of individuals without thyroid dysfunction, visible or palpable goiter, treatment with drugs which can alter thyroid function test results such as glucocorticoids, adrenal insufficiency, renal insufficiency (failure) or other serious systemic diseases, or pregnancy was 10,446. Five tests for thyroid function (FT3, FT4, TSH, TPOAb, and TGAb) were conducted and fasting venous blood samples were collected between 8:00‐10:00 am, revealing that 5345 individuals, aged between 19 and 100 years, who met the criteria of being negative for TPOAb and TGAb antibodies. 

2.Comment: we kindly ask you to copyedit the manuscript in order to improve the English and gain in clarity. Please also note that we have included a copy of the two reviews on your revised version 2 of the manuscript, so please include the changes suggested by the Referee. 

Response: We have made some corrections in language expression marked in red in the “Revised Manuscript with Track Changes”. We improved the clarity of picture. And the manuscript include changes suggested by the Referee. 

3. Reviewer #2 Comment: Are all patients asked to attend in a fasting state and between 0800 and 1000 hours or were data collected only for participants who fitted this criteria?

Response: We choosed the data collected only for participants who fitted in a fasting state and between 8:00‐10:00 am.

Thank you for your consideration. We look forward to hearing from you soon.

If you have any questions, please feel free to contact us.

Best regards

Sincerely,

Liang Xiong

Department of Laboratory Medicine, Liyuan Hospital, Tongji Medical College, Huazhong University of Science and Technology, Yanhu Road, 39 Wuhan, Hubei, PR China

Phone number: 86-027-86785006

Fax number: 86-027-86793043

xionglianghust@163.com

---

## [Editor Report · Decision Letter 3]

10 Sep 2020

Study of reference intervals for free triiodothyronine, free thyroxine, and thyroid-stimulating hormone in an elderly Chinese Han population

PONE-D-19-14834R3

Dear Dr. Xiong,

We’re pleased to inform you that your manuscript has been judged scientifically suitable for publication and will be formally accepted for publication once it meets all outstanding technical requirements.

Kind regards,

Silvia Naitza

Academic Editor

PLOS ONE

Additional Editor Comments (optional):

Dear Dr. Jingting Xiong,

thank you for submitting the revised version of your manuscript PONE-D-19-14834R3 and sorry once again for the delay it took for the entire revision process. I'm pleased to let you know that as it stands now your manuscript can be accepted for publication in this Journal. However, for clarity I suggest you to change the text (row 87-90) as it follows: "Five tests for thyroid function (FT3, FT4,TSH, TPOAb, and TGAb) were conducted in venous blood samples from individuals in a fasting state (collected between 8:00‐10:00 am), revealing that 5345 individuals, aged between 19 and 100 years, met the criteria of being negative for TPOAb and TGAb antibodies".

Thank you again for your patience and best regards,

Silvia Naitza
---

## [Editor Report · Acceptance letter]

14 Sep 2020

PONE-D-19-14834R3 

Study of reference intervals for free triiodothyronine, free thyroxine, and thyroid-stimulating hormone in an elderly Chinese Han population 

Dear Dr. Xiong:

I'm pleased to inform you that your manuscript has been deemed suitable for publication in PLOS ONE. Congratulations! Your manuscript is now with our production department. 

Kind regards, 

on behalf of

Dr. Silvia Naitza 

Academic Editor

PLOS ONE